# Multi-Resolution Learning with DeepONets and Long Short-Term Memory Neural Networks

## Abstract

Deep operator networks (DeepONets, DONs) offer a distinct advantage over traditional neural networks in their ability to be trained on multi-resolution data. This property becomes especially relevant in real-world scenarios where high-resolution measurements are difficult to obtain, while low-resolution data is more readily available. Nevertheless, DeepONets alone often struggle to capture and maintain dependencies over long sequences compared to other state-of-the-art algorithms. We propose a novel framework that leverages multi-resolution data in training and provides precise models when dealing with limited high-resolution data. We achieve this through extending the DeepONet architecture with a long short-term memory network (LSTM), coined DON-LSTM, and training it in a three-step procedure that utilizes data of different levels of granularity. Combining these two architectures, we equip the network with explicit mechanisms to leverage multi-resolution data, as well as capture temporal dependencies in long sequences. We test our method on long-time-evolution modeling of multiple nonlinear systems and show that the proposed multi-resolution DON-LSTM achieves significantly lower generalization error and requires fewer high-resolution samples compared to its vanilla counterparts.

## Introduction

Modeling the temporal evolution of dynamical systems is of paramount importance across various scientific and engineering domains, including physics, biology, climate science, and industrial predictive maintenance. Creating such models enables future predictions and process optimization, and provides insights into the underlying mechanisms governing these systems.

Nevertheless, obtaining precise and comprehensive data for modeling poses a challenge in practical contexts. Real-world scenarios bring along the issue of data limitations, where the costs associated with obtaining large scale, high-resolution data are often prohibitive, e.g., due to specific equipment or increased time required to obtain high-resolution measurements, significant computational costs to perform high-fidelity simulations, or even limited bandwidth of satellite observations. Moreover, historical data may be maintained with reduced granularity due to storage considerations, while in other scenarios, legacy data may be available after an outdated technology is replaced, e.g., resulting in vast amounts of low-resolution sensor data but only limited amounts of high-resolution data.

In such cases, data originating from various sources with differing levels of granularity presents both a challenge and opportunity. While low-resolution data may lack the precision required to learn high-resolution models, it can be exploited by deep learning models to capture general patterns and dynamics that govern the system at hand, and increase the representativeness of the training sample. Multi-resolution learning algorithms serve as a solution for effectively integrating information from data collected over diverse temporal and spatial scales. Beyond addressing data availability constraints, multi-resolution methods can offer a reduction of computational costs during training, initially leveraging only lower-resolution data to capture general dynamics, and subsequently fine-tuning the models on high-resolution data.

To this end, several approaches have been proposed to tackle multi-resolution, multi-scale, or discretization-invariant learning. Among these, a natural framework that has gained prominence

is neural operators, which are neural networks that learn mappings between function spaces, e.g., DeepONet, Fourier neural operator and others (Lu et al., 2021; Li et al., 2020; Cao et al., 2023; Wang & Golland, 2022; Ronneberger et al., 2015; Seidman et al., 2022). Alternative methods for multi-resolution learning include encoder-decoder-based architectures (Ong et al., 2022; Aboutalebi et al., 2022), graph and message-passing networks (Equer et al., 2023; Liu et al., 2021), and combinations of the above (Yang et al., 2022). In a related vein, another body of research deals with conceptually similar multi-fidelity learning in DeepONets, which tackles the problem of combining data of varying quality, where high-quality samples are sparse (Lu et al., 2022b; Howard et al., 2022; De et al., 2023).

In this work, we approach multi-resolution learning through the innate discretization-invariance property of DeepONets. We further propose a new architecture, DON-LSTM, which extends the architecture of the DeepONet with a long short-term memory network (LSTM), in order to capture temporal patterns in sequential outputs of the DeepONet. The main purpose of combining these two architectures is to leverage data of different resolutions in training, effectively increasing the feasible training set, as well as to assist the modeling of time-dependent evolution through explicit mechanisms of the LSTM.

The remainder of this paper is structured as follows. First, we formulate the learning problem at hand. Next, we describe our proposed architecture and the training procedure. Finally, we present and discuss our experimental results on four non-linear partial differential equations (PDEs) and low- and high-resolution training sets of various sizes. Our main findings show that our proposed multi-resolution DON-LSTM achieves lower generalization error than its single-resolution counterparts, which require a much larger high-resolution sample size in order to achieve similar precision.

## 1 PROBLEM STATEMENT

In this study, we learn the operator $\mathcal{N}$ that defines the evolution of a system over time starting from any given initial condition, i.e.:

$$\mathcal{N} : u(x, t = 0) \rightarrow G(u(y)), \tag{1}$$

where $u(x, t = 0)$ is the function defining the initial condition and $G(u(y))$ is the operator describing the evolution over time for any $y = (x, t)$, where $x$ and $t$ are the spatial and temporal coordinates of the system's trajectory.

In practice, $G(u(y))$ is observed at discretized fixed locations $\{(x_1, t_1), ..., (x_m, t_n)\}$, resulting in a vector $[u(x_1, t_1), ..., u(x_m, t_n)] \in \mathbb{R}^{m \times n}$ where $m$ is the total number of spatial discretization points and $n$ is the total number of temporal discretization points that define the full trajectory. For the purpose of this study, we assume that for each system we have two datasets:

- High-resolution set $D_H$ with $N_H$ samples of time resolution $\Delta t_H$,
- Low-resolution set $D_L$ of size $N_L$ samples of time resolution $\Delta t_L$.

For the four considered PDE-based examples in this work, we set $N_H = 4 \times N_L$ and $\Delta t_L = 5 \times \Delta t_H$, while the spatial discretization is the same in both datasets. The multi-resolution network is trained on both datasets. The sizes of the training data are dependent on the complexity of the problem.

## 2 PROPOSED MULTI-RESOLUTION DON-LSTM (OURS)

The schematic representation of the proposed architecture is shown in Figure 1. The architecture consists of a DeepONet followed by a reshaping layer, an LSTM layer, and a final feed-forward layer which brings the output back to the predefined spatial dimension of the solution. The architecture is set up such that the DeepONet outputs are fed as inputs to the LSTM. The DeepONet approximates the solution at locations that are determined by its inputs, which enables the initial discretization-invariant training. During the training of the LSTM, we impose the sequential nature of the outputs through these inputs and use the LSTM to process them as temporal sequences.

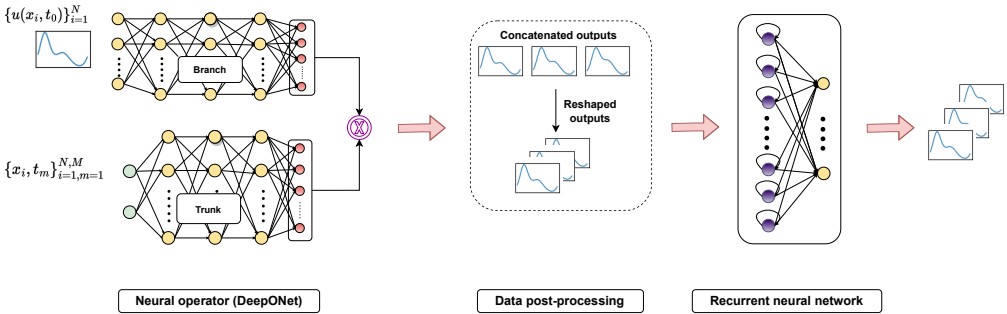

Figure 1: The proposed architecture. In the first phase, the DeepONet maps the solution operator from the initial condition $u_{t=0}$ to solutions at later timesteps. During the DeepONet training, the output $u$ can be defined at arbitrary temporal locations. In the next two training stages, the outputs must have a fixed time interval $\Delta t$ to be compatible with the LSTM. The DeepONet output solutions are reshaped to be represented in a temporally sequential manner and processed by an LSTM. The LSTM lifts the dimension to the specified number of neurons and returns all hidden states. The last layer is a fully-connected dense layer bringing the embedding to the original size of the solution.

## 2.1 DEEP OPERATOR NETWORK (DEEPONET)

The DeepONet is an architecture based on the universal approximation theorem for operators (Chen & Chen, 1995) which combines the outputs of two deep neural networks (the "branch" and "trunk" networks). It is formulated as (Lu et al., 2021):

$$G(u)(y) \approx \sum_{k=1}^{p} \underbrace{b_k(u(x_1), u(x_2), \ldots, u(x_m))}_{\text{branch}} \odot \underbrace{t_k(y)}_{\text{trunk}} , \qquad (2)$$

where $b_k$ and $t_k$ denote the output embeddings of the branch and the trunk network, respectively, and $\odot$ denotes the element-wise multiplication of the outputs of the two networks. The inputs to the branch network are the initial conditions, $u(t = 0)$, discretized at $m$ spatial sensor locations $\{x_1, x_2, \ldots, x_m\}$, and for the trunk network they are the locations, $y = (x, t)$, at which the operator $G(u)$ is evaluated. Additionally, the trunk network can incorporate periodic boundary conditions through simple feature expansion that consists of applying Fourier basis on the provided spatial locations, i.e., $x \to \cos(\frac{2\pi x}{P})$, and $x \to \sin(\frac{2\pi x}{P})$, where $P$ is the period (Lu et al., 2022a).

## 2.2 LONG SHORT-TERM MEMORY NETWORK (LSTM)

LSTM is a type of recurrent neural network (RNN) that was designed to maintain short-term dependencies over long sequences. RNNs in general are a class of specialized neural networks for sequence processing. RNNs employ a hidden state, which persists as the network iterates through data sequences and propagates the information from the previous states forward. LSTMs were developed as an extension to RNNs, aimed at mitigating the problem of vanishing gradients encountered in RNNs. Vanishing gradients occur in long sequences, as the hidden state is updated at every timestep, which leads to a large number of multiplications causing the gradients to approach zero. To address this, LSTM is equipped with a set of gates: the *input gate* regulates the inflow of new information to the maintained hidden state, the *forget gate* determines what proportion of the information should be discarded, and the *output gate* controls the amount of information carried forward.

The choice of the LSTM was especially suitable for our study as our training data were numerical PDE solutions integrated over long time, meaning that each consecutive solution was fully determined by its preceding state. Moreover, we opted to use the LSTM to limit susceptibility to error accumulation, since it directly approximates longer sequences, as opposed to one- or few-steps-ahead prediction methods (Zeng et al., 2023).

## 2.3 SELF-ADAPTIVE LOSS FUNCTION

In solving time-dependent PDEs, the main challenge lies in preserving the precision of the modeled solution over a long-time horizon. Introducing appropriately structured non-uniform penalization parameters can address this aspect. In all the experiments, we equip the networks with self-adaptive weights in the loss function, first introduced by McClenny & Braga-Neto (2020), which adapt during training and force the network to focus on the most challenging regions of the solution. We take the inspiration from Kontolati et al. (2022), where in turn self-adaptive weights considerably improve the accuracy prediction of discontinuities or non-smooth features in the solution. The self-adaptive weights are defined for every spatial and temporal location, and they are updated along with the network parameters during optimization. Specifically, the training loss is defined as:

$$\mathcal{L}(\boldsymbol{\theta}, \boldsymbol{\lambda}) = \frac{1}{N} \sum_{i=1}^{N} g(\boldsymbol{\lambda}) |u(x_i, t_i) - \hat{u}(x_i, t_i)|^2, \tag{3}$$

where $(u(x_i, t_i) - \hat{u}(x_i, t_i))^2$ is the squared error between the reference $u$ and predicted value $\hat{u}$, and $g(\lambda)$ is a non-negative, strictly increasing self-adaptive mask function, and $\boldsymbol{\lambda} = \{\lambda_1, \lambda_2, \cdots \lambda_j\}$ are $j$ self-adaptive parameters, and $j$ is the total number of evaluation points. Typically, in a neural network, we minimize the loss function with respect to the network parameters, $\boldsymbol{\theta}$. However, in this approach, we additionally maximize the loss function with respect to the trainable hyper-parameters using a gradient descent/ascent procedure. The modified objective function is defined as:

$$\min_{\boldsymbol{\theta}} \max_{\boldsymbol{\lambda}} \mathcal{L}(\boldsymbol{\theta}, \boldsymbol{\lambda}). \tag{4}$$

The self-adaptive weights are updated using the gradient descent method, such that

$$\boldsymbol{\lambda}^{k+1} = \lambda^k + \eta_\lambda \nabla_{\boldsymbol{\lambda}} \mathcal{L}(\boldsymbol{\theta}, \boldsymbol{\lambda}), \tag{5}$$

where $\eta_{\boldsymbol{\lambda}}$ is the learning rate of the self-adaptive weights and

$$\nabla_{\lambda_i} \mathcal{L} = \left[ g'(\lambda_i)(u_i(\xi) - \mathcal{G}_\theta(\mathbf{v}_i)(\xi))^2 \right]^T. \tag{6}$$

Therefore, if $g(\lambda_i) > 0$, $\nabla_{\lambda_i} \mathcal{L}$ would be zero only if the term $(u_i(\xi) - \mathcal{G}_\theta(\mathbf{v}_i)(\xi))$ is zero. In this work, the self-adaptive weights are normalized to sum up to one after each epoch. The weights $\lambda$ are updated alongside the weights of the network during training.

## 2.4 TRAINING PROCEDURE

The training procedure was defined with the strategy to first incorporate the low-resolution data in pre-training, capturing the general system's dynamics and increasing the representative training sample, followed by the use of the limited high-resolution data to fine-tune the weights to achieve high-resolution prediction.

In our approach, we propose a three-step procedure:

1. First, we exploit the information provided by the low-resolution data with the DeepONet. This step can be seen as pre-training during which the model captures the general patterns governing the modelled system, while providing the model with a more representative sample of the overall data distribution, which may not be available at the target, high resolution.

2. Next, we extend this network with the mechanisms suitable for the underlying data (here, the LSTM mechanisms to capture short-term dependencies in long sequences). During this phase, we tune the weights of the LSTM with the high-resolution data to increase the model's precision. At the same time, we keep the weights of the DeepONet unchanged, which prevents the LSTM from overfitting to the small high-resolution sample and ensures that the DeepONet preserves the general system's dynamics.

3. Finally, we gently fine-tune the whole architecture with a small learning rate. This step adjusts the DeepONet weights to potentially also capture some of the finer-grained characteristics of the system.

Training is performed iteratively over a predefined number of epochs and in minibatches, i.e., in each epoch the training data is divided into multiple smaller subsets. After calculating the loss on each batch, the gradients are calculated w.r.t. the loss function and the trainable network parameters (weights) are updated. The optimization is performed using the Adam optimizer and a predefined learning rate. The training procedure is also described in Algorithm 1.

---

**Algorithm 1** DON-LSTM training procedure.

---

1: **Step 1:** DEEPONET TRAINING($\boldsymbol{\theta}_{DON}, D_L, lr_1, n_{freq}$)
2:     Initialize $\boldsymbol{\theta}_{DON}$.
3:     Iteratively update $\boldsymbol{\theta}_{DON}$ on $D_L$ with learning rate $lr_1$, save at every $n_{freq}$ epochs.
4:     Compute MSE loss on predictions for validation data.
5:     Choose $\boldsymbol{\theta}_{DON_i}$ with the lowest loss.
6: **Step 2:** LSTM TRAINING($\boldsymbol{\theta}_{LSTM}, \boldsymbol{\theta}_{DON}, D_H, lr_1, n_{freq}$)
7:     Extend DeepONet with reshaping and LSTM layers.
8:     Initialize $\boldsymbol{\theta}_{LSTM}$.
9:     Freeze $\boldsymbol{\theta}_{DON}$ as non-trainable.
10:     Iteratively update $\boldsymbol{\theta}_{LSTM}$ on $D_H$ with learning rate $lr_1$, save at every $n_{freq}$ epochs.
11:     Compute MSE loss on predictions for validation data.
12:     Choose $\boldsymbol{\theta}_{LSTM}$ with the lowest loss.
13: **Step 3:** DON-LSTM TRAINING($\boldsymbol{\theta}_{LSTM}, \boldsymbol{\theta}_{DON}, D_H, lr_2, n_{freq}$)
14:     Unfreeze $\boldsymbol{\theta}_{DON}$ as trainable.
15:     Iteratively update $\boldsymbol{\theta}_{LSTM}$ on $D_H$ with learning rate $lr_2 < lr_1$, save at every $n_{freq}$ epochs.
16:     Compute MSE loss on predictions for validation data.
17:     Choose $\boldsymbol{\theta}_{LSTM}$ with the lowest loss.

---

## 3 PROBLEMS CONSIDERED AND DATA GENERATION

The performance of the proposed multi-resolution models is showcased on four infinite-dimensional non-linear dynamical systems. The numerical solutions to the Korteweg–de Vries, Benjamin–Bona–Mahony and Cahn–Hilliard equations were obtained through schemes that are second order in both space and time; the equations are spatially discretized using central finite differences and integrated in time using the implicit midpoint method. The solutions of the Burgers' equation were generated using PDEBench (Takamoto et al., 2022). All PDEs are evaluated on the one-dimensional spatial domain $\Omega = [0, P]$, for different $P$, with periodic boundary conditions $u(P, t) = u(0, t)$ for all $t \geq 0$. The training set for each PDE consists of data obtained from $N = N_H + N_L$ different initial conditions integrated over time $t = T$ with step size $\Delta t$. The PDEs are described in Appendix A, where example data are visualized in Figure 4, and details of the time and space domains are given in Table 2.

## 4 EXPERIMENTAL RESULTS

We compare the performance of our multi-resolution DON-LSTM against five benchmark networks discussed in section 3 using the relative squared error (RSE), the mean average error (MAE) and the root mean squared error (RMSE), described in Appendix E. The average values of these error metrics are presented in Table 1, and the log values of RSE against the increasing number of high-resolution training samples are shown in Figure 2. Each of the six models is tested on five random weight initializations, four to five different training data sizes, and four problems. Due to space considerations, detailed tables with the evaluation of the models trained on distinct sample sizes are available in Appendix B. All evaluations are performed on high-resolution test samples of size $N_{test} = 1000$.

### 4.1 BENCHMARK MODELS

The benchmark models used to evaluate the performance of the multi-resolution DON-LSTM (DON-LSTM ($D_H, D_L$)) are the following:

- **DON ($D_L$):** The vanilla DeepONet trained on $N_L$ low-resolution data,

- **DON ($D_H$):** The vanilla DeepONet trained on $N_H$ high-resolution data,

- **DON ($D_H, D_L$):** The vanilla DeepONet trained on $N_L + N_H$ multi-resolution data (both datasets),

- **LSTM** ($D_H$): An architecture trained on $N_H$ high-resolution data, consisting of one dense layer lifting the input to a dimension [-1, $mn$], a reshaping layer (into [-1, $m$, $n$]) followed by an LSTM layer, where $m$ is the spatial and $t$ is the temporal dimension,
- **DON-LSTM** ($D_H$): The proposed architecture trained only on $N_H$ high-resolution data.

The models vary by the amount and granularity of the training data. We take advantage of the discretization-invariance of deep neural operators and train the vanilla DeepONet and DON-LSTM on multi-resolution data (in the case of DON-LSTM only the DeepONet layers are trained with multi-resolution). In this problem formulation, the LSTM implicitly learns $\Delta t$ and therefore has to be trained at the resolution used in testing (here, high-resolution). In the following, we refer to specific models by the data resolution used in their training and the models' name, e.g., *high-resolution DON*. In all experiments we chose fixed hyperparameter values and equipped all the models with equal mechanisms that facilitate learning (e.g., self-adaptive weights in the loss function). Moreover, we refrained from fine-tuning the hyperparameters of any particular model.

## 4.2 GENERALIZATION PERFORMANCE

We evaluate the performance of our models grouping them by the number of samples used in their training (Figure 2). As expected, increasing the sample size leads to a reduction in the generalization error for all models. In nearly all cases, we also see that the multi-resolution DON-LSTM achieves the lowest error, followed by the multi-resolution DON for the KdV, BBM and Cahn–Hilliard equations, and the high-resolution LSTM for the Burgers' equation.

Our main findings can be summarized into the following:

1. The multi-resolution DON-LSTM generally achieves the lowest generalization error out of the five benchmarks.

2. In order to achieve similar accuracy with single-resolution methods (such as the vanilla LSTM) we need significantly more high-resolution training samples than for multi-resolution DON-LSTM. For example, in case of the KdV equation, $RSE \approx 0.09$ is achieved by the multi-resolution DON-LSTM with $N_H = 250$, while the LSTM requires $N_H = 750$ (where $N_H$ is the number of high-resolution samples).

3. In multiple cases the DON trained on larger amount of lower-resolution data obtains better results than the DON trained on fewer samples of high-resolution data.

4. While the DeepONet itself achieves reasonable performance, the time-dependent architecture is crucial for capturing long-time dynamics, as is evident by the superior performance of DON-LSTM and the vanilla LSTM.

## 4.3 COMPARISON WITH STATE-OF-THE-ART

To provide a broader understanding of our method's performance, we included the comparison to additional state-of-the-art methods for long time series prediction: a transformer model and a Fourier neural operator (FNO). We employed the FNO variant that contains four Fourier layers ($modes = 8$, $width = 64$) and employs Fourier convolutions through both space and time. The transformer model was composed of a dimension expansion layer (linear mapping), followed by a multi-head attention layer with positional encoding, a layer normalization, and three fully-connected layers.

The multi-resolution DON-LSTM exhibited superior performance over the transformer model on all testing examples, which we attribute to the limited benefit of long-time dependencies for our data. The multi-resolution DON-LSTM also outperformed the FNO tested on the Burgers' and Cahn–Hilliard equations, while the FNO was best on Korteweg–de Vries and Benjamin–Bona–Mahony equations.

## 5 DISCUSSION

In all our experiments, the multi-resolution DON-LSTM achieved the lowest generalization error, while requiring fewer high-resolution training samples than its benchmarks. This advantage stems

| Model | Resolution | | MAE | RMSE | RSE |
|---|---|---|---|---|---|
| **Korteweg–de Vries equation** | | | | | |
| **DON** | $\Delta t = 0.025$ | (high) | 0.190±0.007 | 0.321±0.005 | 0.333±0.010 |
| **DON** | $\Delta t = 0.125$ | (low) | 0.094±0.005 | 0.189±0.006 | 0.125±0.007 |
| **DON** | $\Delta t = \{0.025, 0.125\}$ | (multi) | 0.083±0.005 | 0.175±0.005 | 0.105±0.005 |
| **DON-LSTM** | $\Delta t = 0.025$ | (high) | 0.086±0.013 | 0.168±0.024 | 0.113±0.028 |
| **DON-LSTM** | $\Delta t = \{0.025, 0.125\}$ | (multi) | **0.042±0.002** | 0.122±0.010 | 0.049±0.008 |
| **LSTM** | $\Delta t = 0.025$ | (high) | 0.067±0.003 | 0.200±0.003 | 0.133±0.004 |
| **FNO** | $\Delta t = 0.025$ | (high) | 0.049±0.002 | **0.111±0.005** | **0.042±0.004** |
| **Transformer** | $\Delta t = 0.025$ | (high) | 0.091±0.002 | 0.221±0.006 | 0.170±0.009 |
| **Viscous Burgers' equation** | | | | | |
| **DON** | $\Delta t = 0.01$ | (high) | 0.114±0.001 | 0.168±0.002 | 0.070±0.001 |
| **DON** | $\Delta t = 0.05$ | (low) | 0.089±0.001 | 0.132±0.001 | 0.043±0.001 |
| **DON** | $\Delta t = \{0.01, 0.05\}$ | (multi) | 0.087±0.001 | 0.129±0.001 | 0.044±0.001 |
| **DON-LSTM** | $\Delta t = 0.01$ | (high) | 0.111±0.003 | 0.186±0.005 | 0.087±0.004 |
| **DON-LSTM** | $\Delta t = \{0.01, 0.05\}$ | (multi) | **0.049±0.002** | **0.092±0.003** | **0.022±0.001** |
| **LSTM** | $\Delta t = 0.01$ | (high) | 0.059±0.002 | 0.110±0.001 | 0.032±0.001 |
| **FNO** | $\Delta t = 0.01$ | (high) | 0.078±0.002 | 0.118±0.002 | 0.038±0.001 |
| **Transformer** | $\Delta t = 0.01$ | (high) | 0.133±0.004 | 0.210±0.005 | 0.112±0.005 |
| **Benjamin–Bona–Mahony equation** | | | | | |
| **DON** | $\Delta t = 0.075$ | (high) | 0.278±0.014 | 0.533±0.016 | 0.118±0.007 |
| **DON** | $\Delta t = 0.375$ | (low) | 0.091±0.006 | 0.227±0.007 | 0.021±0.001 |
| **DON** | $\Delta t = \{0.075, 0.375\}$ | (multi) | 0.077±0.007 | 0.191±0.011 | 0.016±0.002 |
| **DON-LSTM** | $\Delta t = 0.075$ | (high) | 0.104±0.023 | 0.264±0.056 | 0.033±0.013 |
| **DON-LSTM** | $\Delta t = \{0.075, 0.375\}$ | (multi) | **0.045±0.003** | 0.151±0.022 | 0.010±0.003 |
| **LSTM** | $\Delta t = 0.075$ | (high) | 0.107±0.010 | 0.332±0.012 | 0.044±0.003 |
| **FNO** | $\Delta t = 0.075$ | (high) | 0.054±0.004 | **0.100±0.007** | **0.005±0.001** |
| **Transformer** | $\Delta t = 0.075$ | (high) | 0.130±0.006 | 0.355±0.015 | 0.057±0.004 |
| **Cahn–Hilliard equation** | | | | | |
| **DON** | $\Delta t = 0.02$ | (high) | 0.041±0.002 | 0.068±0.002 | 0.018±0.001 |
| **DON** | $\Delta t = 0.1$ | (low) | 0.038±0.003 | 0.056±0.004 | 0.013±0.002 |
| **DON** | $\Delta t = \{0.02, 0.1\}$ | (multi) | 0.018±0.001 | 0.031±0.002 | 0.004±0.001 |
| **DON-LSTM** | $\Delta t = 0.02$ | (high) | 0.076±0.009 | 0.137±0.014 | 0.075±0.015 |
| **DON-LSTM** | $\Delta t = \{0.02, 0.1\}$ | (multi) | **0.014±0.001** | **0.027±0.002** | **0.003±0.001** |
| **LSTM** | $\Delta t = 0.02$ | (high) | 0.016±0.001 | 0.036±0.001 | 0.005±0.000 |
| **FNO** | $\Delta t = 0.02$ | (high) | 0.022±0.001 | 0.037±0.001 | 0.005±0.000 |
| **Transformer** | $\Delta t = 0.02$ | (high) | 0.029±0.001 | 0.061±0.001 | 0.015±0.000 |

Table 1: The mean and the standard deviation of the prediction errors of all models. Each reported value aggregates the mean error values across all models trained on different sample sizes, where each model has been trained five times (i.e., the mean and standard deviation of the values reported in Appendix B).

from the combination of two factors: the utilization of a larger training dataset, which encompasses both high- and low-resolution samples, and the integration of LSTM mechanisms that facilitate capturing the temporal evolution of the systems. The inclusion of low-resolution data in early training contributes to the improvement of the prediction, as seen in the superior performance of multi-resolution DON-LSTM as compared to the vanilla LSTM and single-resolution DON-LSTM, as well as the superior performance of the multi-resolution DON in comparison to both single-resolution DONs. As evidenced by this result, the LSTM and DON-LSTM networks are not able to effectively capture the mechanisms that govern the data without the low-resolution data pre-training.

Additionally, we specifically observe that the inclusion of LSTM mechanisms is beneficial, as evidenced by the multi-resolution DON-LSTM outperforming the vanilla DON trained on multi-resolution data.

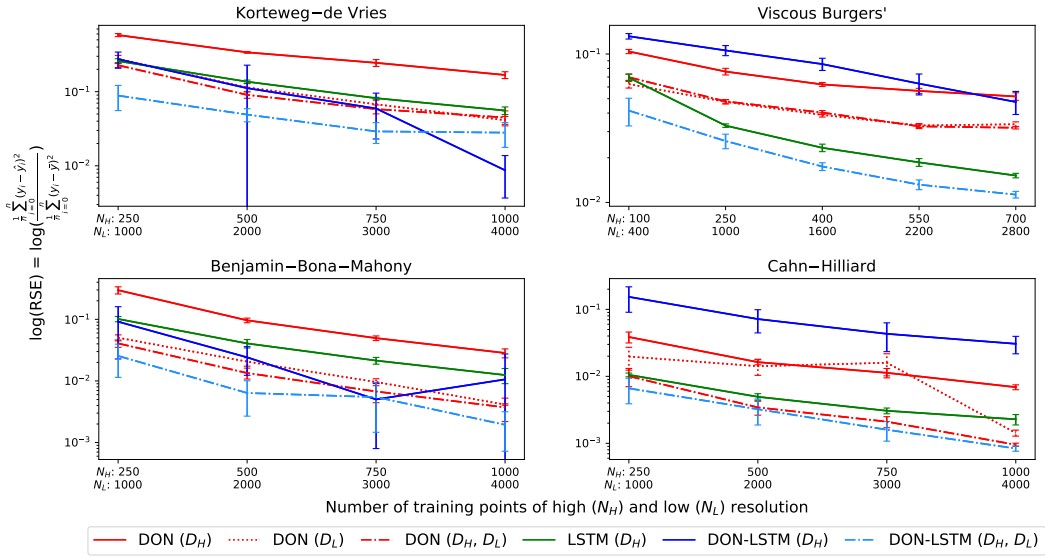

Figure 2: Model performance vs. amount of training samples. The $y$-axis shows the relative squared error (log values) for predictions on high-resolution test data, while the $x$-axis indicates the number of high- and low-resolution training samples. The figure compares the performance of three models: DON (red) and DON-LSTM (blue, with light-blue for multi-resolution) and LSTM (green), with variations of training data denoted by $D_\square$ and the line pattern ($D_H$ and solid line for high-resolution, $D_L$ and dotted line for low-resolution, and $D_H$, $D_L$ and dash-dotted line for multi-resolution). The error bars indicate the $95\%$ confidence interval calculated over the five trained models (two standard deviations of the error). See Appendix B, Figure 5 for the absolute RSE values.

When low-resolution data was not used in training, the comparison between the architectures was inconclusive, i.e., the high-resolution DON-LSTM outperformed its vanilla counterparts only in two experiments. We attribute it to the fact that DON-LSTM is comprised of a larger number of parameters, and a small training sample is not sufficient to effectively train the network, leading to under/over-fitting. We can also see that the vanilla DON struggles with adjusting all its parameters on a small sample, which becomes apparent through the fact that the model trained on a low-resolution data achieves better performance than when trained on the high-resolution data (regardless of being tested on high-resolution). This means that the inclusion of low-resolution data in early training is essential for the good performance of the proposed architecture.

## 6 LIMITATIONS AND FUTURE WORK

While DeepONets possess the discretization-invariance property in the output function, they require the input data to be defined at fixed locations. This problem is addressed in the literature through the employment of an encoder-decoder architecture integrated with the DeepONet (Ong et al., 2022; Zhang et al., 2022). We also note that the framework that is limited to discretization-invariant output is sufficient for applications where the system behaviour is modeled from a single-resolution input, e.g., an initial condition. For cases when multi-resolution input is desired, we highlight the existence of other neural operator architectures such as the Fourier neural operator (Li et al., 2020), or the Laplace neural operator (Cao et al., 2023).

In addition, we note that LSTM is specifically suited and limited to sequential data, best capturing short-term dependencies. We see this as an opportunity for future studies, in which the DeepONet can be extended with architectures appropriate for different types of data, for example convolutional neural networks in case of image data, or transformers for data governed by long-term dependencies and global temporal patterns.

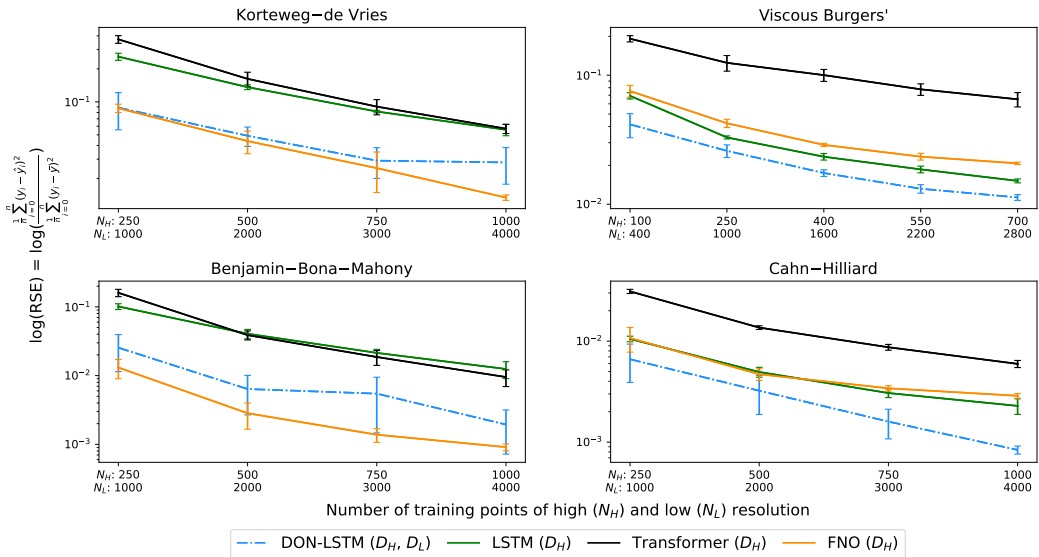

Figure 3: Model performance vs. amount of training samples. The $y$-axis shows the relative squared error (log values) for predictions on high-resolution test data, while the $x$-axis indicates the number of high- and low-resolution training samples. The figure compares the performance of four models: multi-resolution DON-LSTM (blue), LSTM (green), transformer (black), and FNO (orange), with variations of training data denoted by $D_\square$ and the line pattern ($D_H$ and solid line for high-resolution, and $D_H$, $D_L$ and dash-dotted line for multi-resolution). The error bars indicate the $95\%$ confidence interval calculated over the five trained models (two standard deviations of the error). See Appendix B, Figure 6 for the absolute RSE values.

## 7 CONCLUSIONS

Our proposed multi-resolution framework seamlessly integrates the strengths of its two composites: the discretization invariance of the DeepONet and the improved sequential learning with the memory-preserving mechanisms of the LSTM. We have demonstrated that these properties can be leveraged to incorporate multi-resolution data in training, as well as to capture the intricate dynamics of time-dependent systems, leading to significantly improved predictive accuracy in modeling of the systems' evolution over long-time horizons. Our experiments clearly demonstrate the efficacy of our approach in creating accurate, high-resolution models even with limited training data available at fine-grained resolution. Moreover, the synergistic effect of our proposed architecture makes it an apt choice for real-world scenarios, promising substantial enhancements in prediction quality. This work not only advances the understanding and utilization of multi-resolution data in sequential analysis but also provides valuable insights for future research and applications.

## 8 REPRODUCIBILITY

The code for reproducing the results is available in anonymous GitHub repository. The code includes the default parameters to generate the models and the data processing pipeline used in this paper. The details of the used architectures, training setup and data processing steps are also specified in Appendix C and Appendix D.

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

## A  PARTIAL DIFFERENTIAL EQUATIONS DATA

### A.1  KORTEWEG–DE VRIES EQUATION

The Korteweg–de Vries (KdV) equation (Korteweg & De Vries, 1895) is a non-linear dispersive PDE that describes the evolution of small-amplitude, long-wavelength systems in a variety of physical settings, such as shallow water waves, ion-acoustic waves in plasmas, and certain types of nonlinear optical waves. We consider a one-dimensional unforced case, which is given by:

$$\frac{\partial u}{\partial t} + \gamma \frac{\partial^3 u}{\partial x^3} - \eta u \frac{\partial u}{\partial x} = 0, \tag{7}$$

where $u := u(x, t)$ is the height of the wave at position $x$ at time $t$, and $\eta = 6$ and $\gamma = 1$ are chosen real-valued scalar parameters.

The initial conditions are each a sum of two solitons (solitary waves), *i.e.*:

$$u(x, 0) = \sum_{i=1}^{2} 2k_i^2 \text{sech}^2 \left( k_i((x + \frac{P}{2} - Pd_i)\%P - \frac{P}{2}) \right), \tag{8}$$

where sech stands for hyperbolic secant ($\frac{1}{\cosh}$), $P$ is the period in space, $\%$ is the modulo operator, $i = \{1, 2\}$, $k_i \in (0.5, 1.0)$ and $d_i \in (0, 1)$ are coefficients that determine the height and location of the peak of a soliton, respectively. For each initial state $t = 0$ in the training data, these coefficients are drawn randomly from their distribution.

### A.2  BENJAMIN–BONA–MAHONY EQUATION

The Benjamin–Bona–Mahony (BBM) equation was derived as a higher-order improvement on the KdV equation and includes both non-linear and dispersive effects (Peregrine, 1966; Benjamin et al., 1972). It is used for studying a broader range of wave behaviors, including wave breaking and dispersion, and is given by:

$$\frac{\partial u}{\partial t} - \frac{\partial^3 u}{\partial x^2 \partial t} + \frac{\partial}{\partial x} \left( u + \frac{1}{2} u^2 \right) = 0. \tag{9}$$

The chosen initial conditions are superpositions of two soliton waves of the following shape:

$$u_i(x, 0) = \sum_{i=1}^{2} 3(c_i - 1)\text{sech}^2 \left( \frac{1}{2} \sqrt{1 - \frac{1}{c_i}} (x + \frac{P}{2} - Pd_i)\%P - \frac{P}{2} \right), \tag{10}$$

where $c_i \in (1, 3)$ and $d_i \in (0, 1)$ are coefficients that determine the height and location of the peak of a soliton, respectively.

### A.3  CAHN–HILLIARD EQUATION

The Cahn–Hilliard equation is used to describe phase separation with applications to materials science and physics (Cahn & Hilliard, 1958). It is expressed as:

$$\frac{\partial u}{\partial t} - \frac{\partial^2}{\partial x^2}(\nu u + \alpha u^3 + \mu \frac{\partial^2 u}{\partial x^2}) = 0, \tag{11}$$

where we set $\nu = -0.01$, $\alpha = 0.01$ and $\mu = -0.00001$.

The initial conditions are superpositions of sine and cosine waves, i.e. $u(x, 0) = u_1 + u_2$ with:

$$u_i(x, 0) = a_i \sin(k_i \frac{2\pi}{P} x) + b_i \cos(j_i \frac{2\pi}{P} x), \tag{12}$$

where $a_i, b_i \in (0, 0.2)$ and $k_i, j_i$ are integers and $j_i, k_i \in [1, 6]$.

## A.4 VISCOUS BURGERS' EQUATION

The viscous Burgers' equation (Bateman, 1915; Burgers, 1948) describes the behavior of waves and shocks in a viscous fluid or gas. It is given by:

$$\frac{\partial u}{\partial t} + \frac{\partial}{\partial x}\left(\frac{u^2}{2}\right) = \frac{\nu}{\pi}\frac{\partial^2 u}{\partial^2 x}, \tag{13}$$

where $x \in (0,1)$, $t \in (0,2]$, and $\nu = 0.001$ is the diffusion coefficient.

The initial conditions are given by the superposition of sinusoidal waves:

$$u(x,0) = \sum_{k_i = k_1, \ldots, k_N} A_i \sin(k_i x + \phi_i), \tag{14}$$

where $k_i = 2\pi n_i / P$ are coefficients where $n_i$ are arbitrarily selected integers in $[1, n_{max}]$. $N$ is the integer determining how many waves are added, $A_i$ is a random float number uniformly chosen in $(0,1)$, and $\phi_i$ is the randomly chosen phase in $(0, 2\pi)$.

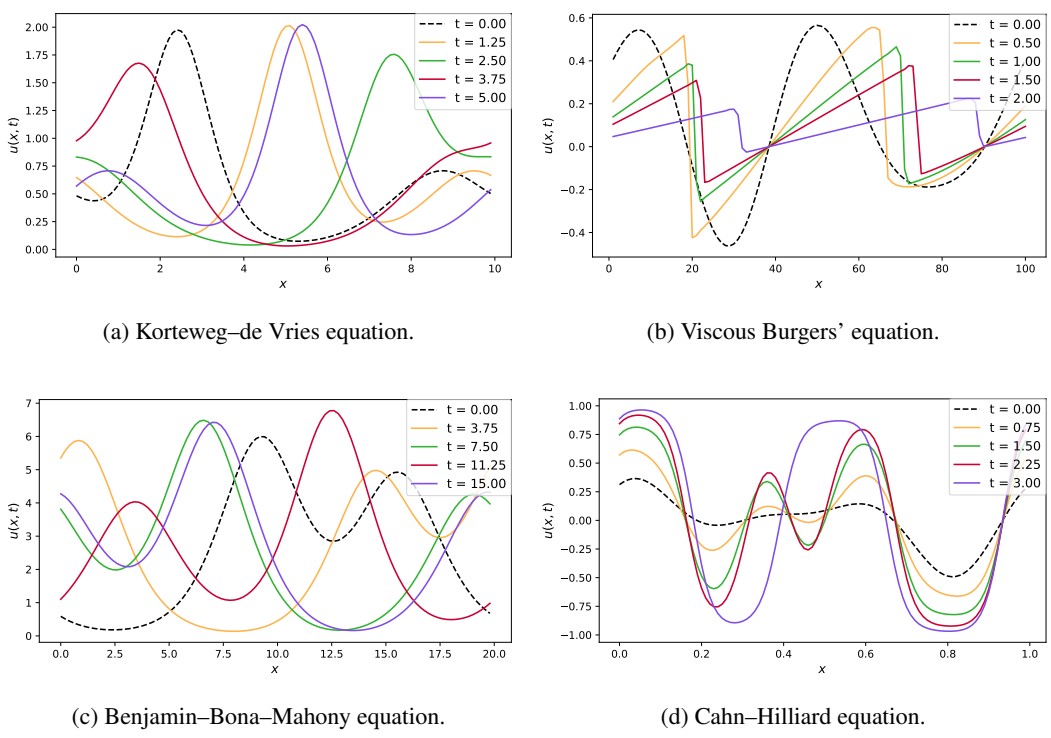

(a) Korteweg–de Vries equation.

(b) Viscous Burgers' equation.

(c) Benjamin–Bona–Mahony equation.

(d) Cahn–Hilliard equation.

Figure 4: Random samples from training data. Each dataset consists of multiple trajectories starting from different initial conditions.

| Equation | $N$ | $t_H$ | $t_L$ | $x$ |
|---|---|---|---|---|
| Korteweg–de Vries | 10000 | • $t \in [0, 5]$
• $\Delta t = 0.025$
• 201 points | • $t \in [0, 5]$
• $\Delta t = 0.125$
• 41 points | • $\Omega \in [0, 10]$
• $\Delta x = 0.1$
• 100 points |
| Viscous Burgers' | 5000 | • $t \in [0, 2]$
• $\Delta t = 0.01$
• 201 points | • $t \in [0, 2]$
• $\Delta t = 0.05$
• 41 points | • $\Omega \in [-1, 1]$
• $\Delta x = 0.02$
• 100 points |
| Benjamin–Bona–Mahony | 10000 | • $t \in [0, 15]$
• $\Delta t = 0.075$
• 201 points | • $t \in [0, 15]$
• $\Delta t = 0.375$
• 41 points | • $\Omega \in [0, 20]$
• $\Delta x = 0.2$
• 100 points |
| Cahn–Hilliard | 10000 | • $t \in [0, 3]$
• $\Delta t = 0.02$
• 151 points | • $t \in [0, 3]$
• $\Delta t = 0.1$
• 31 points | • $\Omega \in [0, 1]$
• $\Delta x = 0.01$
• 100 points |

Table 2: Data available: data size, i.e. number of samples ($N$), time domain of high-resolution ($t_H$) and low-resolution data ($t_L$), spatial domain ($x$). We only use a small subset of this data in training.

## B  GENERALIZATION PERFORMANCE

**Korteweg–de Vries equation**

| Model | Resolution | | MAE | RMSE | RSE |
|---|---|---|---|---|---|
| $N_H = 250, N_L = 1000$ | | | | | |
| **DON** | $\Delta t = 0.025$ | (high) | 0.298±0.003 | 0.434±0.005 | 0.580±0.014 |
| **DON** | $\Delta t = 0.125$ | (low) | 0.179±0.008 | 0.300±0.008 | 0.277±0.016 |
| **DON** | $\Delta t = \{0.025, 0.125\}$ | (multi) | 0.149±0.012 | 0.273±0.005 | 0.227±0.008 |
| **DON-LSTM** | $\Delta t = 0.025$ | (high) | 0.152±0.011 | 0.298±0.018 | 0.274±0.034 |
| **DON-LSTM** | $\Delta t = \{0.025, 0.125\}$ | (multi) | **0.065±0.005** | 0.169±0.016 | 0.089±0.016 |
| **LSTM** | $\Delta t = 0.025$ | (high) | 0.112±0.006 | 0.290±0.005 | 0.259±0.010 |
| **FNO** | $\Delta t = 0.025$ | (high) | 0.081±0.002 | **0.169±0.004** | **0.088±0.004** |
| **Transformer** | $\Delta t = 0.025$ | (high) | 0.164±0.002 | 0.348±0.007 | 0.372±0.015 |
| $N_H = 500, N_L = 2000$ | | | | | |
| **DON** | $\Delta t = 0.025$ | (high) | 0.192±0.011 | 0.332±0.002 | 0.339±0.005 |
| **DON** | $\Delta t = 0.125$ | (low) | 0.084±0.002 | 0.193±0.006 | 0.114±0.007 |
| **DON** | $\Delta t = \{0.025, 0.125\}$ | (multi) | 0.075±0.004 | 0.169±0.004 | 0.091±0.005 |
| **DON-LSTM** | $\Delta t = 0.025$ | (high) | 0.089±0.025 | 0.185±0.050 | 0.112±0.058 |
| **DON-LSTM** | $\Delta t = \{0.025, 0.125\}$ | (multi) | **0.044±0.002** | 0.126±0.006 | 0.049±0.005 |
| **LSTM** | $\Delta t = 0.025$ | (high) | 0.069±0.003 | 0.211±0.003 | 0.136±0.003 |
| **FNO** | $\Delta t = 0.025$ | (high) | 0.048±0.001 | **0.119±0.007** | **0.044±0.005** |
| **Transformer** | $\Delta t = 0.025$ | (high) | 0.089±0.003 | 0.230±0.009 | 0.162±0.012 |
| $N_H = 750, N_L = 3000$ | | | | | |
| **DON** | $\Delta t = 0.025$ | (high) | 0.149±0.004 | 0.282±0.008 | 0.245±0.013 |
| **DON** | $\Delta t = 0.125$ | (low) | 0.066±0.005 | 0.148±0.005 | 0.068±0.004 |
| **DON** | $\Delta t = \{0.025, 0.125\}$ | (multi) | 0.059±0.003 | 0.137±0.005 | 0.058±0.004 |
| **DON-LSTM** | $\Delta t = 0.025$ | (high) | 0.077±0.010 | 0.138±0.021 | 0.059±0.018 |
| **DON-LSTM** | $\Delta t = \{0.025, 0.125\}$ | (multi) | **0.034±0.002** | 0.097±0.008 | 0.029±0.005 |
| **LSTM** | $\Delta t = 0.025$ | (high) | 0.050±0.001 | 0.163±0.001 | 0.082±0.001 |
| **FNO** | $\Delta t = 0.025$ | (high) | 0.037±0.002 | **0.090±0.009** | **0.025±0.005** |
| **Transformer** | $\Delta t = 0.025$ | (high) | 0.064±0.002 | 0.171±0.007 | 0.091±0.007 |
| $N_H = 1000, N_L = 4000$ | | | | | |
| **DON** | $\Delta t = 0.025$ | (high) | 0.123±0.010 | 0.234±0.006 | 0.168±0.009 |
| **DON** | $\Delta t = 0.125$ | (low) | 0.047±0.006 | 0.116±0.004 | 0.041±0.003 |
| **DON** | $\Delta t = \{0.025, 0.125\}$ | (multi) | 0.048±0.002 | 0.121±0.007 | 0.045±0.005 |
| **DON-LSTM** | $\Delta t = 0.025$ | (high) | 0.024±0.004 | **0.053±0.008** | **0.009±0.003** |
| **DON-LSTM** | $\Delta t = \{0.025, 0.125\}$ | (multi) | **0.023±0.001** | 0.096±0.009 | 0.028±0.005 |
| **LSTM** | $\Delta t = 0.025$ | (high) | 0.039±0.003 | 0.135±0.004 | 0.056±0.003 |
| **FNO** | $\Delta t = 0.025$ | (high) | 0.030±0.001 | 0.066±0.001 | 0.013±0.000 |
| **Transformer** | $\Delta t = 0.025$ | (high) | 0.047±0.001 | 0.136±0.003 | 0.057±0.003 |

Table 3: The mean and the standard deviation of prediction metrics on high-resolution data. Each model is trained 5 times.

**Viscous Burgers' equation ($\nu = 0.001$)**

| Model | Resolution | | MAE | RMSE | RSE |
|---|---|---|---|---|---|
| $N_H = 100, N_L = 400$ | | | | | |
| **DON** | $\Delta t = 0.01$ | (high) | 0.139±0.001 | 0.207±0.002 | 0.104±0.002 |
| **DON** | $\Delta t = 0.05$ | (low) | 0.109±0.001 | 0.161±0.002 | 0.063±0.002 |
| **DON** | $\Delta t = \{0.01, 0.05\}$ | (multi) | 0.108±0.001 | 0.158±0.002 | 0.070±0.002 |
| **DON-LSTM** | $\Delta t = 0.01$ | (high) | 0.157±0.001 | 0.233±0.002 | 0.132±0.003 |
| **DON-LSTM** | $\Delta t = \{0.01, 0.05\}$ | (multi) | **0.076±0.005** | **0.131±0.007** | **0.042±0.004** |
| **LSTM** | $\Delta t = 0.01$ | (high) | 0.106±0.003 | 0.169±0.002 | 0.069±0.002 |
| **FNO** | $\Delta t = 0.01$ | (high) | 0.121±0.004 | 0.171±0.005 | 0.075±0.004 |
| **Transformer** | $\Delta t = 0.01$ | (high) | 0.184±0.003 | 0.281±0.004 | 0.192±0.005 |
| $N_H = 250, N_L = 1000$ | | | | | |
| **DON** | $\Delta t = 0.01$ | (high) | 0.120±0.002 | 0.177±0.002 | 0.076±0.002 |
| **DON** | $\Delta t = 0.05$ | (low) | 0.094±0.001 | 0.140±0.001 | 0.048±0.001 |
| **DON** | $\Delta t = \{0.01, 0.05\}$ | (multi) | 0.092±0.001 | 0.138±0.001 | 0.048±0.001 |
| **DON-LSTM** | $\Delta t = 0.01$ | (high) | 0.132±0.002 | 0.208±0.004 | 0.106±0.004 |
| **DON-LSTM** | $\Delta t = \{0.01, 0.05\}$ | (multi) | **0.055±0.003** | **0.103±0.003** | **0.026±0.001** |
| **LSTM** | $\Delta t = 0.01$ | (high) | 0.062±0.002 | 0.116±0.001 | 0.033±0.000 |
| **FNO** | $\Delta t = 0.01$ | (high) | 0.087±0.002 | 0.129±0.002 | 0.042±0.002 |
| **Transformer** | $\Delta t = 0.01$ | (high) | 0.142±0.006 | 0.226±0.008 | 0.125±0.009 |
| $N_H = 400, N_L = 1600$ | | | | | |
| **DON** | $\Delta t = 0.01$ | (high) | 0.109±0.001 | 0.160±0.001 | 0.062±0.001 |
| **DON** | $\Delta t = 0.05$ | (low) | 0.085±0.001 | 0.126±0.001 | 0.039±0.001 |
| **DON** | $\Delta t = \{0.01, 0.05\}$ | (multi) | 0.083±0.001 | 0.124±0.001 | 0.040±0.001 |
| **DON-LSTM** | $\Delta t = 0.01$ | (high) | 0.108±0.003 | 0.187±0.004 | 0.086±0.004 |
| **DON-LSTM** | $\Delta t = \{0.01, 0.05\}$ | (multi) | **0.044±0.001** | **0.085±0.001** | **0.017±0.001** |
| **LSTM** | $\Delta t = 0.01$ | (high) | 0.050±0.001 | 0.098±0.001 | 0.023±0.001 |
| **FNO** | $\Delta t = 0.01$ | (high) | 0.068±0.001 | 0.106±0.001 | 0.029±0.000 |
| **Transformer** | $\Delta t = 0.01$ | (high) | 0.127±0.002 | 0.203±0.005 | 0.100±0.005 |
| $N_H = 550, N_L = 2200$ | | | | | |
| **DON** | $\Delta t = 0.01$ | (high) | 0.104±0.001 | 0.152±0.002 | 0.056±0.001 |
| **DON** | $\Delta t = 0.05$ | (low) | 0.077±0.001 | 0.116±0.001 | 0.033±0.000 |
| **DON** | $\Delta t = \{0.01, 0.05\}$ | (multi) | 0.075±0.001 | 0.114±0.001 | 0.033±0.001 |
| **DON-LSTM** | $\Delta t = 0.01$ | (high) | 0.087±0.004 | 0.161±0.007 | 0.063±0.005 |
| **DON-LSTM** | $\Delta t = \{0.01, 0.05\}$ | (multi) | **0.037±0.001** | **0.074±0.001** | **0.013±0.000** |
| **LSTM** | $\Delta t = 0.01$ | (high) | 0.042±0.002 | 0.087±0.001 | 0.019±0.001 |
| **FNO** | $\Delta t = 0.01$ | (high) | 0.059±0.001 | 0.096±0.001 | 0.023±0.001 |
| **Transformer** | $\Delta t = 0.01$ | (high) | 0.112±0.002 | 0.179±0.005 | 0.078±0.004 |
| $N_H = 700, N_L = 2800$ | | | | | |
| **DON** | $\Delta t = 0.01$ | (high) | 0.099±0.001 | 0.146±0.002 | 0.052±0.002 |
| **DON** | $\Delta t = 0.05$ | (low) | 0.078±0.001 | 0.118±0.001 | 0.034±0.001 |
| **DON** | $\Delta t = \{0.01, 0.05\}$ | (multi) | 0.075±0.000 | 0.113±0.001 | 0.032±0.000 |
| **DON-LSTM** | $\Delta t = 0.01$ | (high) | 0.071±0.003 | 0.140±0.006 | 0.048±0.004 |
| **DON-LSTM** | $\Delta t = \{0.01, 0.05\}$ | (multi) | **0.033±0.000** | **0.068±0.001** | **0.011±0.000** |
| **LSTM** | $\Delta t = 0.01$ | (high) | 0.036±0.001 | 0.079±0.001 | 0.015±0.000 |
| **FNO** | $\Delta t = 0.01$ | (high) | 0.055±0.000 | 0.090±0.000 | 0.021±0.000 |
| **Transformer** | $\Delta t = 0.01$ | (high) | 0.101±0.005 | 0.163±0.005 | 0.065±0.004 |

Table 4: The mean and the standard deviation of prediction metrics on high-resolution data. Each model is trained 5 times.

**Benjamin–Bona–Mahony**

| Model | Resolution | | MAE | RMSE | RSE |
|---|---|---|---|---|---|
| $N_H = 250, N_L = 1000$ | | | | | |
| **DON** | $\Delta t = 0.075$ | (high) | 0.554±0.019 | 0.932±0.031 | 0.297±0.020 |
| **DON** | $\Delta t = 0.375$ | (low) | 0.166±0.012 | 0.383±0.011 | 0.050±0.003 |
| **DON** | $\Delta t = \{0.075, 0.375\}$ | (multi) | 0.155±0.011 | 0.331±0.011 | 0.041±0.003 |
| **DON-LSTM** | $\Delta t = 0.075$ | (high) | 0.198±0.041 | 0.510±0.095 | 0.092±0.034 |
| **DON-LSTM** | $\Delta t = \{0.075, 0.375\}$ | (multi) | **0.088±0.005** | 0.271±0.036 | 0.025±0.007 |
| **LSTM** | $\Delta t = 0.075$ | (high) | 0.191±0.008 | 0.544±0.013 | 0.101±0.005 |
| **FNO** | $\Delta t = 0.075$ | (high) | 0.098±0.006 | **0.195±0.015** | **0.013±0.002** |
| **Transformer** | $\Delta t = 0.075$ | (high) | 0.264±0.006 | 0.684±0.021 | 0.160±0.010 |
| $N_H = 500, N_L = 2000$ | | | | | |
| **DON** | $\Delta t = 0.075$ | (high) | 0.250±0.010 | 0.530±0.011 | 0.096±0.004 |
| **DON** | $\Delta t = 0.375$ | (low) | 0.096±0.006 | 0.245±0.010 | 0.021±0.002 |
| **DON** | $\Delta t = \{0.075, 0.375\}$ | (multi) | 0.074±0.008 | 0.192±0.010 | 0.013±0.001 |
| **DON-LSTM** | $\Delta t = 0.075$ | (high) | 0.107±0.020 | 0.264±0.033 | 0.024±0.006 |
| **DON-LSTM** | $\Delta t = \{0.075, 0.375\}$ | (multi) | **0.042±0.002** | 0.135±0.020 | 0.006±0.002 |
| **LSTM** | $\Delta t = 0.075$ | (high) | 0.111±0.018 | 0.344±0.014 | 0.041±0.003 |
| **FNO** | $\Delta t = 0.075$ | (high) | 0.050±0.006 | **0.091±0.009** | **0.003±0.001** |
| **Transformer** | $\Delta t = 0.075$ | (high) | 0.117±0.006 | 0.337±0.013 | 0.039±0.003 |
| $N_H = 750, N_L = 3000$ | | | | | |
| **DON** | $\Delta t = 0.075$ | (high) | 0.181±0.018 | 0.380±0.008 | 0.050±0.002 |
| **DON** | $\Delta t = 0.375$ | (low) | 0.058±0.006 | 0.168±0.005 | 0.010±0.001 |
| **DON** | $\Delta t = \{0.075, 0.375\}$ | (multi) | 0.046±0.002 | 0.140±0.012 | 0.007±0.001 |
| **DON-LSTM** | $\Delta t = 0.075$ | (high) | 0.048±0.007 | 0.119±0.025 | 0.005±0.002 |
| **DON-LSTM** | $\Delta t = \{0.075, 0.375\}$ | (multi) | **0.032±0.006** | 0.125±0.022 | 0.005±0.002 |
| **LSTM** | $\Delta t = 0.075$ | (high) | 0.068±0.005 | 0.250±0.008 | 0.021±0.001 |
| **FNO** | $\Delta t = 0.075$ | (high) | 0.037±0.002 | **0.063±0.004** | **0.001±0.000** |
| **Transformer** | $\Delta t = 0.075$ | (high) | 0.081±0.009 | 0.233±0.015 | 0.019±0.002 |
| $N_H = 1000, N_L = 4000$ | | | | | |
| **DON** | $\Delta t = 0.075$ | (high) | 0.125±0.009 | 0.288±0.012 | 0.028±0.002 |
| **DON** | $\Delta t = 0.375$ | (low) | 0.043±0.002 | 0.110±0.001 | 0.004±0.000 |
| **DON** | $\Delta t = \{0.075, 0.375\}$ | (multi) | 0.033±0.006 | 0.103±0.011 | 0.004±0.001 |
| **DON-LSTM** | $\Delta t = 0.075$ | (high) | 0.061±0.023 | 0.162±0.071 | 0.011±0.008 |
| **DON-LSTM** | $\Delta t = \{0.075, 0.375\}$ | (multi) | **0.019±0.001** | 0.074±0.011 | 0.002±0.001 |
| **LSTM** | $\Delta t = 0.075$ | (high) | 0.056±0.008 | 0.191±0.013 | 0.012±0.002 |
| **FNO** | $\Delta t = 0.075$ | (high) | 0.030±0.001 | **0.052±0.001** | **0.001±0.000** |
| **Transformer** | $\Delta t = 0.075$ | (high) | 0.057±0.004 | 0.167±0.012 | 0.010±0.001 |

Table 5: The mean and the standard deviation of prediction metrics on high-resolution data. Each model is trained 5 times.

**Cahn–Hilliard equation**

| Model | Resolution | | MAE | RMSE | RSE |
|---|---|---|---|---|---|
| $N_H = 250, N_L = 1000$ | | | | | |
| **DON** | $\Delta t = 0.02$ | (high) | 0.066±0.004 | 0.104±0.005 | 0.039±0.004 |
| **DON** | $\Delta t = 0.1$ | (low) | 0.050±0.006 | 0.074±0.007 | 0.020±0.004 |
| **DON** | $\Delta t = \{0.02, 0.1\}$ | (multi) | 0.031±0.003 | 0.053±0.004 | 0.010±0.002 |
| **DON-LSTM** | $\Delta t = 0.02$ | (high) | 0.125±0.015 | 0.206±0.021 | 0.154±0.032 |
| **DON-LSTM** | $\Delta t = \{0.02, 0.1\}$ | (multi) | **0.023±0.003** | **0.043±0.004** | **0.007±0.001** |
| **LSTM** | $\Delta t = 0.02$ | (high) | 0.025±0.000 | 0.054±0.001 | 0.011±0.000 |
| **FNO** | $\Delta t = 0.02$ | (high) | 0.032±0.002 | 0.054±0.004 | 0.011±0.001 |
| **Transformer** | $\Delta t = 0.02$ | (high) | 0.049±0.001 | 0.093±0.001 | 0.031±0.001 |
| $N_H = 500, N_L = 2000$ | | | | | |
| **DON** | $\Delta t = 0.02$ | (high) | 0.041±0.001 | 0.067±0.001 | 0.016±0.001 |
| **DON** | $\Delta t = 0.1$ | (low) | 0.044±0.002 | 0.063±0.004 | 0.014±0.002 |
| **DON** | $\Delta t = \{0.02, 0.1\}$ | (multi) | 0.018±0.001 | 0.031±0.002 | **0.003±0.000** |
| **DON-LSTM** | $\Delta t = 0.02$ | (high) | 0.079±0.010 | 0.141±0.013 | 0.072±0.014 |
| **DON-LSTM** | $\Delta t = \{0.02, 0.1\}$ | (multi) | **0.016±0.002** | **0.030±0.003** | 0.003±0.001 |
| **LSTM** | $\Delta t = 0.02$ | (high) | 0.015±0.000 | 0.037±0.001 | 0.005±0.000 |
| **FNO** | $\Delta t = 0.02$ | (high) | 0.022±0.001 | 0.036±0.001 | 0.005±0.000 |
| **Transformer** | $\Delta t = 0.02$ | (high) | 0.028±0.002 | 0.062±0.001 | 0.014±0.000 |
| $N_H = 750, N_L = 3000$ | | | | | |
| **DON** | $\Delta t = 0.02$ | (high) | 0.034±0.002 | 0.056±0.002 | 0.011±0.001 |
| **DON** | $\Delta t = 0.1$ | (low) | 0.047±0.004 | 0.066±0.006 | 0.016±0.003 |
| **DON** | $\Delta t = \{0.02, 0.1\}$ | (multi) | 0.013±0.001 | 0.024±0.001 | 0.002±0.000 |
| **DON-LSTM** | $\Delta t = 0.02$ | (high) | 0.055±0.007 | 0.109±0.013 | 0.043±0.010 |
| **DON-LSTM** | $\Delta t = \{0.02, 0.1\}$ | (multi) | **0.011±0.001** | **0.021±0.002** | **0.002±0.000** |
| **LSTM** | $\Delta t = 0.02$ | (high) | 0.013±0.001 | 0.029±0.001 | 0.003±0.000 |
| **FNO** | $\Delta t = 0.02$ | (high) | 0.018±0.000 | 0.031±0.000 | 0.003±0.000 |
| **Transformer** | $\Delta t = 0.02$ | (high) | 0.021±0.001 | 0.049±0.001 | 0.009±0.000 |
| $N_H = 1000, N_L = 4000$ | | | | | |
| **DON** | $\Delta t = 0.02$ | (high) | 0.025±0.002 | 0.044±0.001 | 0.007±0.000 |
| **DON** | $\Delta t = 0.1$ | (low) | 0.010±0.000 | 0.020±0.001 | 0.001±0.000 |
| **DON** | $\Delta t = \{0.02, 0.1\}$ | (multi) | 0.009±0.000 | 0.016±0.000 | 0.001±0.000 |
| **DON-LSTM** | $\Delta t = 0.02$ | (high) | 0.045±0.003 | 0.092±0.007 | 0.031±0.004 |
| **DON-LSTM** | $\Delta t = \{0.02, 0.1\}$ | (multi) | **0.007±0.000** | **0.015±0.000** | **0.001±0.000** |
| **LSTM** | $\Delta t = 0.02$ | (high) | 0.011±0.001 | 0.025±0.001 | 0.002±0.000 |
| **FNO** | $\Delta t = 0.02$ | (high) | 0.017±0.000 | 0.028±0.000 | 0.003±0.000 |
| **Transformer** | $\Delta t = 0.02$ | (high) | 0.016±0.000 | 0.041±0.001 | 0.006±0.000 |

Table 6: The mean and the standard deviation of prediction metrics on high-resolution data. Each model is trained 5 times.

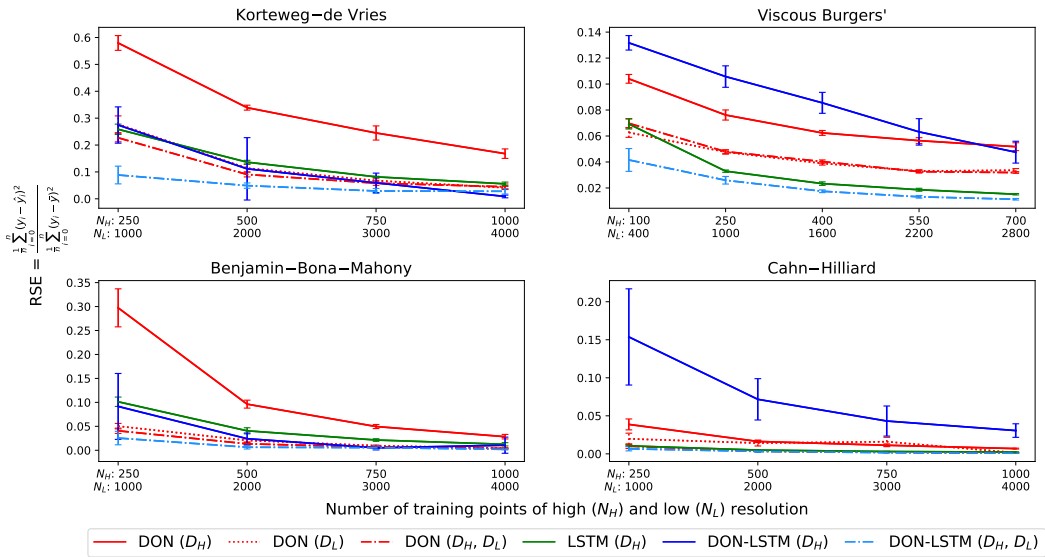

Figure 5: Model performance vs. amount of training samples. The $y$-axis shows the relative squared error for predictions on high-resolution test data, while the $x$-axis indicates the number of high- and low-resolution training samples. The figure compares the performance of three models: DON (red) and DON-LSTM (blue, with light-blue for multi-resolution) and LSTM (green), with variations of training data denoted by $D_\square$ and the line pattern ($D_H$ and solid line for high-resolution, $D_L$ and dotted line for low-resolution, and $D_H$, $D_L$ and dash-dotted line for multi-resolution). The error bars indicate the 95% confidence interval calculated over the five trained models (two standard deviations of the error).

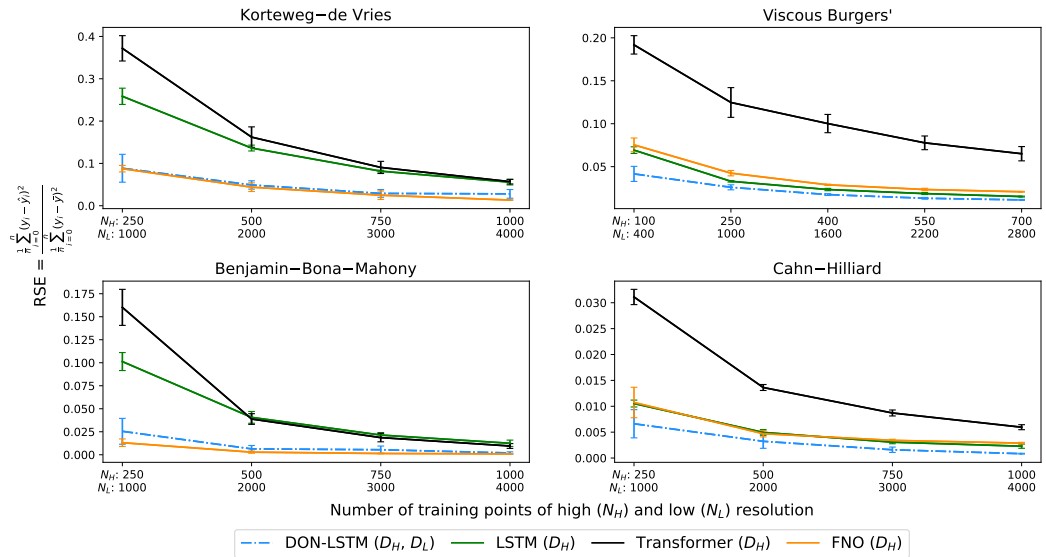

Figure 6: Model performance vs. amount of training samples. The $y$-axis shows the relative squared error for predictions on high-resolution test data, while the $x$-axis indicates the number of high- and low-resolution training samples. The figure compares the performance of four models: multi-resolution DON-LSTM (blue), LSTM (green), transformer (black), and FNO (orange), with variations of training data denoted by $D_\square$ and the line pattern ($D_H$ and solid line for high-resolution, and $D_H, D_L$ and dash-dotted line for multi-resolution). The error bars indicate the $95\%$ confidence interval calculated over the five trained models (two standard deviations of the error).

## C ARCHITECTURE DETAILS

### C.1 DEEPONET

The vanilla DeepONet is constructed with a branch and trunk network with the same output sizes, which are merged using Einstein summation (Figure 1). The input to the branch network is of the size [batch_size, x_len], the input to the trunk network is of the size [x_len×t_len, 2], and the output of the network after the Einstein summation is [batch_size, x_len×t_len], where batch_size refers to the number of samples in each minibatch, and x_len and t_len, to the sizes of the spatial and temporal dimensions, *i.e.*, the number of discretization points.

Table 7: Branch network.

| | LAYER | OUTPUT SHAPE | ACTIVATION | PARAM |
|---|---|---|---|---|
| 0 | INPUT | (NONE, X) | - | 0 |
| 1 | DENSE | (NONE, 150) | SWISH | $150x + 150$ |
| 2 | DENSE | (NONE, 250) | SWISH | $37,750$ |
| 3 | DENSE | (NONE, 450) | SWISH | $112,950$ |
| 4 | DENSE | (NONE, 380) | SWISH | $171,380$ |
| 5 | DENSE | (NONE, 320) | SWISH | $121,920$ |
| 6 | DENSE | (NONE, 300) | LINEAR | $96,300$ |

Table 8: Trunk network.

|   | LAYER | OUTPUT SHAPE | ACTIVATION | PARAM |
|---|-------|--------------|------------|-------|
| 0 | INPUT | (NONE, 2) | - | 0 |
| 1 | DENSE | (NONE, 200) | SWISH | 600 |
| 2 | DENSE | (NONE, 220) | SWISH | 44, 220 |
| 3 | DENSE | (NONE, 240) | SWISH | 53, 040 |
| 4 | DENSE | (NONE, 250) | SWISH | 60, 250 |
| 5 | DENSE | (NONE, 260) | SWISH | 65, 260 |
| 6 | DENSE | (NONE, 280) | LINEAR | 73, 080 |
| 7 | DENSE | (NONE, 300) | LINEAR | 84, 300 |

The DeepONet is trained in minibatches of 50 samples using the Adam optimizer and learning rate of $1e-4$. For the DeepONet the data is normalized in the following manner: the inputs to the branch network and the outputs of the DeepONet use standard scaling, and the inputs to the trunk network use min-max (Appendix D). The vanilla DeepONet for the full trajectory is trained up to $25,000$ epochs.

## C.2 DON-LSTM

Table 9: LSTM extension for the input of $t$ timesteps.

|   | LAYER | OUTPUT SHAPE | ACTIVATION | PARAM |
|---|-------|--------------|------------|-------|
| 0 | DEEPONET OUTPUT | (NONE, $x \times t$) | - | 0 |
| 1 | RESHAPE | (NONE, $t, x$) | - | 0 |
| 2 | LSTM | (NONE, $t, 200$) | TANH | $4 \times ((x+1) \times 200 + 200^2)$ |
| 3 | DENSE | (NONE, $t, x$) | LINEAR | $xt + x$ |

The inputs and outputs in the LSTM training are normalized using standard scaling.

## D   DATA SCALING DETAILS

When training the DeepONets, the inputs to the branch net are scaled using standard scaling, while the inputs to the trunk net use the min-max scaling. For the LSTM, the inputs are scaled using standard scaling. The outputs are always scaled with the standard scaling.

### D.1   STANDARD SCALING

The standard scaling formula is defined as:

$$x' = \frac{x - \mu}{\sigma},$$  (15)

where $x'$ are the standardized values, $x$ are the original values, $\mu$ is the mean, and $\sigma$ is the standard deviation of $x$.

### D.2   MIN-MAX SCALING

Min-max scaling, also known as Min-max normalization, scales the data between 0 and 1. It is expressed by the equation:

$$x' = \frac{x - x_{min}}{x_{max} - x_{min}},$$  (16)

where $x'$ are the normalized values, $x$ are the original values, and $x_{min}$ and $x_{max}$ are the minimal and maximal values of $x$.

# E  EVALUATION METRICS

In this study, we used several metrics to evaluate the performance of the model such as mean average error, root mean squared error, and relative squared error.

## E.1  MEAN AVERAGE ERROR

The mean average error (MAE) is expressed as:

$$\text{MAE} = \frac{1}{n} \sum_{i=1}^{n} |y_i - \hat{y}_i|, \tag{17}$$

where $n$ is the number of samples, $y_i$ is the true value of the $i^{th}$ sample, and $\hat{y}_i$ is the predicted value of the $i^{th}$ sample.

## E.2  ROOT MEAN SQUARED ERROR

The root mean squared error (RMSE) is expressed as:

$$\text{RMSE} = \sqrt{\frac{1}{n} \sum_{i=1}^{n} (y_i - \hat{y}_i)^2}, \tag{18}$$

where: $n$ is the number of samples, $y_i$ is the true value of the $i^{th}$ sample and $\hat{y}_i$ is the predicted value of the $i^{th}$ sample.

## E.3  RELATIVE SQUARED ERROR

The relative squared error (RSE) is the total squared error between the predicted values and the ground truth normalized by the total squared error between the ground truth and the mean. RSE is interpreted on the scale between 0-1, where 0 indicates the perfect fit, while values of 1 and larger are obtained only if the model's prediction is worse than fitting the mean line. The metric is expressed as:

$$\text{RSE} = \frac{\frac{1}{n} \sum_{i=1}^{n} (y_i - \hat{y}_i)^2}{\frac{1}{n} \sum_{i=1}^{n} (y_i - \bar{y})^2} \tag{19}$$

where $y_i$ is the true value of the $i^{th}$ sample, $\hat{y}_i$ is the predicted value of the $i^{th}$ sample, and $\bar{y}$ is the mean value of all samples.

