# OpenReview forum: "Multi-Resolution Learning with DeepONets and Long Short-Term Memory Neural Networks"
_ICLR.cc/2024/Conference — Submitted to ICLR 2024_

### Official Review · Reviewer_FcVp · 2023-10-30

**Soundness:** 2 fair
**Presentation:** 3 good
**Contribution:** 1 poor
**Rating:** 3
**Confidence:** 4

**Summary:**

Deep operator networks (DeepONets or DONs) have a unique capability to be trained on multi-resolution data, a significant advantage in real-world contexts where high-resolution data may be challenging to acquire. However, traditional DeepONets face difficulties in maintaining dependencies over extended sequences. To address this, the paper introduces a novel architecture called DON-LSTM, which merges the benefits of DeepONets with the temporal pattern recognition of long short-term memory networks (LSTM). This combination allows the model to effectively utilize multi-resolution data and capture time-dependent evolutions. The newly proposed DON-LSTM aims to harness both multi-resolution data and temporal patterns, improving the predictive accuracy for long-time system evolutions. Results indicate that this architecture offers lower generalization errors than considered baselines.

**Strengths:**

* The problem is well motivated and the paper is nicely structured
* This particular combination of LSTMs and DeepONets has not been done before
* Code is submitted

**Weaknesses:**

* The proposed model completely lacks novelty. It is simply a combination of DeepONets (which have already been around for several years) and the most prominent RNN architecture LSTM (which has been around for several decades).

* The experimental results section is very weak. The considered baselines are not interesting nor meaningful. The paper should compare their results with other competing method that have been shown to perform well on these problems. In particular the paper should compare their results with FNOs (and their variants), and standard CNNs.

* The obtained results should be reported in relative errors (normalized by the scale of the problem). This is particularly important in engineering applications. Based on its current form one cannot check if the models obtain 1\% error, 100\% error, or even more.

* No theory is provided.

* A multi-resolution approach appears to be only applied during the training procedure, but is not tested during inference.

* Instead of using LSTMs it would be interesting to use current state-of-the-art RNN architectures that are known to perform well on long-term dependencies.

**Questions:**

* How are the low-resolution data points obtained? Simply a down-scaling of high-resolution data? If so, what is the benefit of training with low-resolution data at all? Is it much faster? Please elaborate on that.

* Why is the training procedure described in 2.4 chosen? Can you provide any ablations on alternative procedures (e.g., permutation of the roles and steps)?

---

> ### Author Response · Authors · 2023-11-20
> **Response to Reviewer FcVp (1/3)**
>
> We want to thank the reviewer for the time and effort put into reviewing our paper. The insightful comments provided us with some invaluable suggestions on improving our paper, which we happily incorporated. Moreover, it was brought to our attention that certain aspects of our communication may not have been as clear as intended. We present the results of the suggested tests, as well as we clarify the questions raised.
>
> ## Contribution and novelty
>
> We want to emphasize that the primary contribution of our work lies in developing a robust framework to successfully leverage multi-resolution data during training, providing precise models when dealing with limited high-resolution data. While we acknowledge the apparent simplicity of our proposed solution, we respectfully disagree with the reviewer’s assessment on the significance of our contribution.
>
> In our approach, we propose a three-step procedure:
> 1. First, we exploit the information provided by the low-resolution data with the deep operator network. This step can be seen as pre-training during which the model captures the general patterns governing the modelled system. Moreover, this step also provides the model with a more representative sample of the overall data distribution, which may not be available at the target, high resolution.
> 2. Next, we extend this network with the mechanisms suitable for the underlying data (here, the LSTM mechanisms to capture short-term dependencies in long sequences). During this phase, we tune the weights of the LSTM with the high-resolution data to increase the model’s precision. At the same time, we keep the weights of the DeepONet unchanged, which prevents the LSTM from overfitting to the small high-resolution sample and ensures that the DeepONet preserves the general system’s dynamics.
> 3. Finally, we gently fine-tune the whole architecture with a small learning rate. This step adjusts the DeepONet weights to also capture some of the finer-grained characteristics of the system.
>
> Consequently, we want to highlight that our contribution is not limited to simply putting two well-validated architectures together but also involves a thoughtful training procedure that leads to a synergistic result:
> 1. The inclusion of the low-resolution data in pre-training allows the model to capture and maintain the general dynamics of the system. With this pre-training, the network can take advantage of the LSTM mechanisms in the next training step and learn a more accurate sequential prediction. This claim is evidenced by the comparisons to the standalone LSTM and the single-resolution DON-LSTMs which perform considerably worse (due to overfitting/the lack of a representative data sample).
> 2. Achieving comparable results by standalone networks, such as the LSTM and the DON requires a significantly higher number of high-resolution samples. For example, in case of the KdV equation, $RSE\approx 0.09$ is achieved by the multi-resolution DON-LSTM with $N_H = 250$, while the LSTM requires $N_H = 750$ (where $N_H$ is the number of high-resolution samples).
>
> Additionally, we want to highlight that our approach is not limited to the architectures explored in the paper. Due to the mentioned simplicity of our method, other combinations of the operators and the state-of-the-art methods depending on the underlying data can be explored: e.g., DON-transformer for multi-resolution learning with global patterns or DON-CNN for image data. We also note that the DeepONet can be replaced with another suitable neural operator.
>
> We trust that these supplementary explanations clarify the novelty and contribution of our research.

---

> ### Author Response · Authors · 2023-11-20
> **Response to Reviewer FcVp (2/3)**
>
> ## Other baselines
>
> We have included the comparison to additional state-of-the-art methods for time series regression: a transformer model and a Fourier neural operator (FNO): https://i.imgur.com/jG6gr9A.png
>
> We employed the FNO variant that contains four Fourier layers (modes=8, width=64) and employs Fourier convolutions through both space and time. The transformer model was composed of a dimension expansion layer (linear mapping), followed by a multi-head attention layer with positional encoding, a layer normalization, and three fully-connected layers.
>
> The multi-resolution DON-LSTM exhibited superior performance over the transformer model on all testing examples, which we attribute to the limited benefit of long-time dependencies for our data. The multi-resolution DON-LSTM also outperformed the FNO tested on the Burgers' and Cahn--Hilliard equations, while the FNO was best on Korteweg--de Vries and Benjamin--Bona--Mahony equations.
>
> We also experimented with convolutional neural networks (CNNs) with convolutions in both space and time, however, this approach led to a notable increase in the generalization error. We attribute this outcome to the nature of our learning problem, where the objective is to map a lower-dimensional input to a significantly higher-dimensional, sequential output (initial conditions defined in space to solutions in time $\times$ space). While CNNs excel in identifying local patterns and capturing hierarchical features, they are primarily effective in applications that reduce or maintain the problem dimension (e.g., image classification, or multivariate sequence to single sequence mapping).
>
> ## Relative errors
>
> All results were reported in the relative squared error (RSE) which is interpreted on the scale between 0-1, where 0 indicates the perfect fit, while values of 1 and larger are obtained only if the model's prediction is worse than fitting the mean line. We find the RSE to be the most suitable and interpretable metric to express the generalization error since it accounts both for the scale and the variation of the data.
>
> For the definition of the relative squared error (RSE), we refer to the Appendix, section E.3:
>
> "The relative squared error (RSE) is the total squared error between the predicted values and the ground truth normalized by the total squared error between the ground truth and the mean. The metric is expressed as:
> \begin{equation}
>     \text{RSE} =
>     \frac{
>         \frac{1}{n}\sum_{i=1}^{n}(y_i - \hat{y_i})^2
>         }
>         {
>         \frac{1}{n} \sum_{i=1}^{n}(y_i - \bar{y})^2
>         }
> \end{equation}
> where $y_i$ is the true value of the $i^{th}$ sample, $\hat{y_i}$ is the predicted value of the $i^{th}$ sample, and $\bar{y}$ is the mean value of all samples."
>
> ## Theory
>
> The architecture of the DeepONet is based on the universal theorem for operators introduced by Chen & Chen in 1995: https://ieeexplore.ieee.org/document/392253
>
> The universal function approximation for neural networks goes back to 1989: https://link.springer.com/article/10.1007/BF02551274, https://www.sciencedirect.com/science/article/pii/0893608089900208
>
> While the universal function approximation theorem may not be directly applicable to LSTMs, their architecture is inspired by traditional neural networks and should hold in certain conditions. The networks are combined in the following way: the DeepONet learns the mapping from the initial condition to the solution in time, then after the DeepONet is extended, the LSTM learns the mapping from the DeepONet’s outputs to the solution in time. Given this sequential approach, the models should complement each other without interference, providing a coherent learning framework.
>
> ## Multi-resolution approach in testing
>
> As noted by the reviewer, all our tests are performed on high-resolution prediction. We intentionally chose high-resolution prediction as our target, since we see limited value in measuring the low-resolution performance: by definition, this would lead to less accurate models. We also note that the high-resolution predictions can be easily downsampled.
>
> ## Exploring long-time dependency architectures
>
> As reported in an earlier response, we tested our models against a transformer model for timeseries data, which is known to capture global patterns and long-term dependencies. However, we note that the LSTM was a natural architecture choice given that our data were numerical PDE solutions governed by local patterns. LSTM is designed to maintain short-term dependencies over long sequences, while in our data each numerical solution is fully determined by its preceding state. For other problems, we agree that other architectures may be more relevant.

---

> ### Author Response · Authors · 2023-11-20
> **Response to Reviewer FcVp (3/3)**
>
> ## Obtaining low-resolution data points
>
> The low-resolution points were obtained through downsampling of the high-resolution data with $\Delta t_L = 5\times \Delta t_H$, where $t_L$ and $t_H$ were the resolution of the low- and high-resolution data, respectively.
>
> Reducing the data resolution simulates many real-world scenarios where high-resolution data is scarce, while low-resolution data is more accessible. For example, high-resolution measurements may be more costly to obtain because they require more sophisticated equipment or increased experimentation time (or more costly simulations). Other scenarios where low-resolution data may be more readily available include limited storage considerations, the legacy data left after replacing an outdated sensor technology, or even limited bandwidth of satellite observations.
>
> In cases when high-resolution data is limited, low-resolution data serves as a useful source of information about general patterns or dynamics that govern the system at hand, even though it lacks the precision required to learn high-resolution models. This ensures that the model is exposed to a more representative sample of the overall distribution, which leads to improved generalization on new, unseen data, and reduces the risk of overfitting to the otherwise limited samples.
>
> Additionally, the training on low-resolution data is indeed faster. For example, training the DeepONet on low-resolution data is 2-3 times faster on the sample of size 1000.
>
> ## Training procedure
>
> The training procedure was defined with the strategy to first incorporate the low-resolution data in pre-training, capturing the general system's dynamics, followed by the use of the (limited) high-resolution data to fine-tune the weights to achieve high-resolution prediction. While we acknowledge the reviewer's suggestion to explore alternative training procedures, we want to highlight that our composed models trained with the current regime consistently outperformed their vanilla baselines. Consequently, we opted to not prioritize optimization of this procedure, and instead we presented the details of the training procedure mostly for clarity and reproducibility. We also explain the reasoning behind the proposed training procedure in the earlier response.

---

### Official Review · Reviewer_qWGe · 2023-11-01

**Soundness:** 3 good
**Presentation:** 4 excellent
**Contribution:** 2 fair
**Rating:** 6
**Confidence:** 3

**Summary:**

The research introduces the multi-resolution DON-LSTM, a novel architecture designed to model time-dependent systems. By merging the strengths of DeepONet's discretization invariance and LSTM's memory-preserving mechanisms, the model leverages both high- and low-resolution training data for improved accuracy. Experimental results demonstrated that as training sample size increased, the generalization error decreased for all models. Notably, the multi-resolution DON-LSTM consistently outperformed benchmarks, achieving the lowest generalization error and requiring fewer high-resolution samples to match the accuracy of single-resolution methods. Key findings include the superior performance of models trained with early-stage low-resolution data and the pivotal role of LSTM mechanisms in enhancing model accuracy. The research also identified potential limitations, emphasizing the need for fixed location input data in DeepONets and suggesting possible solutions like encoder-decoder architectures. Conclusively, the DON-LSTM offers promising advancements in the realm of time-dependent system modeling, highlighting its potential in real-world applications and paving the way for future multi-resolution data studies.

**Strengths:**

The paper presents a new model, the multi-resolution DON-LSTM, that combines two powerful architectures, DeepONet and LSTM, tailored for time-dependent systems.

Authors propose a training mechanism to train the DON-LSTM on both low and high resolution data that leads to better performance.

The paper carries out thorough experimental evaluations against five benchmark models, assessing the proposed architecture. The authors have included the standard errors for the performance obtained.

The utilization of both high- and low-resolution training data in the model allows for enhanced learning, especially when high-resolution samples are limited. The paper offers multiple conclusions from its experiments, such as the superior performance of multi-resolution DON-LSTM over its benchmarks.

Authors have included the limitations and future work suggestion.

**Weaknesses:**

Based on the results, DON-LSTM trained on high resolution data only doesn’t perform better compared to other benchmarks. What extra information does low resolution data provides to the model that leads to increased performance of the DON-LSTM trained on high- and low-resolution data. Also given that LSTM are used in the model, how long sequences can be trained with the model.

The authors can potentially include more baselines to compare their models with. For example, they can include ensemble of DON-LSTM trained on low- and high-resolution data separately or they can also include some other state of art methods used to solve the problem (if they exist)

The authors have described a training mechanism for the DON-LSTM. It would be interesting to analyze, how sensitive the model performance is with respect to training procedure described in the paper. For example, if DeepONet is trained first on high resolution data rather than low resolution data.

Authors have included the standard errors for the loss obtained. However, based on the standard errors, it's hard to conclude if the observed improvements are statistically significant.

Overall, the paper is a smart combination of two different existing architectures to solve a problem and the paper is lacking the theoretical justification regarding the choice of architecture.

**Questions:**

I have listed my concerns and questions in weaknesses section.

---

> ### Author Response · Authors · 2023-11-20
> **Response to Reviewer qWGe (1/2)**
>
> We extend our gratitude to the reviewer for dedicating their time and effort to the comprehensive review of our paper and the constructive feedback provided. We believe that addressing the reviewer's comments will significantly improve our manuscript. The paper will be revised shortly to align with the suggested improvements and clarify the points that were raised.
>
> ## Low-resolution data as additional source of information
>
> Regarding the contribution of low-resolution data to the overall performance of our models: Low-resolution data serves as a useful source of information about general patterns or dynamics that govern the system at hand, even though it lacks the precision required to learn high-resolution models. In our experiments, we use low-resolution data in pre-training, before we transition to the target high-resolution data. The expansion of the dataset ensures that the model is exposed to a more representative sample of the overall distribution, which leads to improved generalization on new, unseen data, and reduces the risk of overfitting to the otherwise limited samples. These factors collectively contribute to the improved predictive performance of the multi-resolution DON-LSTM.
>
> ## Sequence length and training
>
> The reviewer brings up an important question regarding the feasible sequence length during training LSTMs and other sequential models. While we have not explicitly studied the limits of the sequence length, we employed sequences of length 150-200 points, which encapsulated significant changes and the evolution over time within the chosen systems.
>
> Acknowledging a potential limitation on the sequence length, it’s important to emphasize that, regardless of the length that is determined in training, the models can be used recursively during inference. This means that the model's predictions can be provided as new inputs, extending the prediction further in time. Additionally, we highlight that the models trained for sequential mappings are less prone to error accumulation in long-time prediction compared to one- of few-steps-ahead predictions, hence they may be preferred even if the sequence length is limited for a given application (reference: https://ojs.aaai.org/index.php/AAAI/article/view/26317).
>
> ## More baselines
>
> We have included the comparison to additional state-of-the-art methods for time series regression: a transformer model and a Fourier neural operator (FNO): https://i.imgur.com/jG6gr9A.png
>
> We employed the FNO variant that contains four Fourier layers (modes=8, width=64) and employs Fourier convolutions through both space and time. The transformer model was composed of a dimension expansion layer (linear mapping), followed by a multi-head attention layer with positional encoding, a layer normalization, and three fully-connected layers.
>
> The multi-resolution DON-LSTM exhibited superior performance over the transformer model on all testing examples, which we attribute to the limited benefit of long-time dependencies for our data. The multi-resolution DON-LSTM also outperformed the FNO tested on the Burgers' and Cahn--Hilliard equations, while the FNO was best on Korteweg--de Vries and Benjamin--Bona--Mahony equations.
>
> While this comparative analysis provides a broader understanding of our method’s performance, we want to emphasize that the primary contribution of our work lies in developing a robust framework to successfully leverage multi-resolution data in training. Given that our data were numerical PDE solutions, the LSTM was a natural choice: LSTM is designed to maintain short-term dependencies over long sequences, while in our data each numerical solution is fully determined by its preceding state.
>
> Moreover, we suggest considering the combinations with other state-of-the-art methods depending on the underlying data: e.g., DON-transformer for multi-resolution learning with global patterns or DON-CNN for image data. We also note that FNO, as a neural operator, can potentially be used instead of the DeepONet.
>
> Finally, addressing the suggestion of comparing our models to an ensemble of DON-LSTM models trained separately on the low- and high-resolution data, we wish to clarify that since LSTM requires a fixed resolution, a DON-LSTM trained exclusively on low-resolution data would not be applicable to high-resolution inference. Consequently, such such comparison is infeasible within the constraints of the LSTM architecture.

---

> ### Author Response · Authors · 2023-11-20
> **Response to Reviewer qWGe (2/2)**
>
> ## Training procedure
>
> The training procedure was defined with the strategy to first incorporate the low-resolution data in pre-training, capturing the general system's dynamics, followed by the use of the (limited) high-resolution data to fine-tune the weights to achieve high-resolution prediction. While we acknowledge the reviewer's suggestion to explore alternative training procedures, we want to highlight that our composed models trained with the current regime consistently outperformed their vanilla baselines. Consequently, we opted to not prioritize optimization of this procedure, and instead we presented the details of the training procedure mostly for clarity and reproducibility.
>
> As for the suggested permutation of the steps in the training procedure, i.e., first training the DeepONet on the high-resolution and then low-resolution data, we suspect that it would reduce the performance of our models.
>
> ## Statistical significance
>
> We want to thank our reviewer for bringing the issue of statistical significance to our attention. While revising our paper, we noticed that our reported standard deviation in the aggregated table (Table 1) was incorrect. The standard deviation should have been calculated over the models trained on different data samples separately, while we aggregated it over all of them (leading to incorrectly high reported standard deviations).
> We have also modified the color scheme in our plots to enhance their clarity:
>
> The figures of the models' generalization performance can be found below:
>
> log(RSE): https://i.imgur.com/DAKJXOJ.png
>
> RSE: https://i.imgur.com/kcb5J65.png
>
> With this revised report it is clear that the multi-resolution DON-LSTM has consistent, stable performance, while achieving lower generalization error than its vanilla counterparts in nearly all cases. Inspecting the relative squared errors aggregated within equivalent training samples (Appendix B), we can see that the DON-LSTM at its worst has the 95\% confidence interval (two standard deviations) of 0.032.
>
> ## The choice of the architecture
>
> As mentioned in the ealier response, the LSTM was a natural choice for our data, due to its ability to maintain short-term dependencies over long sequences. Our data were numerical PDE solutions integrated over time, where each consecutive solution was fully determined by its preceding solution (exhibiting short-term dependencies). The DeepONet, on the other hand, was chosen due to its ability to learn on multi-resolution data.

---

### Official Review · Reviewer_LP9x · 2023-11-04

**Soundness:** 3 good
**Presentation:** 3 good
**Contribution:** 2 fair
**Rating:** 5
**Confidence:** 2

**Summary:**

Authors propose a new architecture, DON-LSTM that combines the discretization invariance of deep operator networks and the ability of LSTMs to model dependencies in long sequences of multi-resolution data. The authors test their method on various models of non-linear systems, with multi-resolution data  and show improved generalization error, as well as needing fewer high-resolution samples.

**Strengths:**

The core idea of the paper is simple and intuitive, combine the capabilities of DeepONets with LSTMs for more robust modelling of evolving systems. For the various PDEs considered, DON-LSTM performs better than using just naive LSTM or DeepONets.

**Weaknesses:**

The effect of the self-adaptive loss function, in particular the effects of step sizes $\eta_\lambda$ for the gradient ascent step (5) is not discussed. It would be nice to see how this choice effects the stability of the learned operator as well as the general gradient descent convergence behavior.

**Questions:**

While the aggregate errors have been provided, the stability of the learned operator over a long prediction sequence is not demonstrated. Is it possible to provide a figure similar to Figure 3, that shows the predicted values and training data for some initial-conditions?

---

> ### Author Response · Authors · 2023-11-19
> **Response to Reviewer LP9x**
>
> We wish to express our gratitude to the reviewer for dedicating their time and effort to review our paper. We greatly appreciate the received feedback, and it will certainly help us improve our manuscript.
>
> ## Self-adaptive weights, learning rate
>
> Firstly, we want to address the comment about exploring the sensitivity to the learning rate of the self-adaptive weights in the loss function. In all experiments we chose fixed hyperparameter values, which we believe was the fairest way to compare our models. We equipped all the models with equal mechanisms that facilitate learning, such as self-adaptive weights in the loss function, and refrained from fine-tuning any of the hyperparameters of any particular model.
>
> While we agree that experimenting with different hyperparameters could be interesting, we want to highlight that it would make the scope of our study unfeasible: we trained 510 models for this experiment alone (six models, four problems, four to five sample sizes, and five random weight initializations), and the number of explorable hyperparameters is also considerably large (e.g., the learning rate and learning rate schedules, activation functions, number of layers and neurons, different optimizers, batch sizes, weights initializations, weights regularization parameters, etc.).
>
> To specifically address the learning rate $\eta_\lambda$ of the self-adaptive weights, we want to highlight the purpose of employing them: The weights increase for the parts of the solution which the model struggles with and otherwise converge to a unit. Modifying $\eta_\lambda$ mostly influences the rate at which the problematic regions, typically the boundaries or later timesteps, are improved. Since we trained all our models well past the early stopping criterion (25k epochs, while the lowest validation error was typically achieved at 5000-20000 epochs), we believe that changing this hyperparameter would not significantly influence the final precision of our models.
>
> ## Plots of predictions
>
> In below links, we provide the plots of our reference data vs. models’ predictions for all trained models (presented on only one random initial condition each). The dashed black lines are the initial conditions, the solid colored lines are the reference data, and the dotted colored lines are the predictions:
>
> KdV ($N_H=250$, $N_L=1000$): https://i.imgur.com/UfWTaQx.png
>
> KdV ($N_H=500$, $N_L=2000$): https://i.imgur.com/fbtj7jb.png
>
> KdV ($N_H=750$, $N_L=3000$): https://i.imgur.com/dB6MgYw.png
>
> KdV ($N_H=1000$, $N_L=4000$): https://i.imgur.com/nWSC3DO.png
>
> Burgers' ($N_H=100$, $N_L=400$): https://i.imgur.com/AAaATWV.png
>
> Burgers' ($N_H=250$, $N_L=1000$): https://i.imgur.com/vGoe6IA.png
>
> Burgers' ($N_H=400$, $N_L=1600$): https://i.imgur.com/XMKIqCn.png
>
> Burgers' ($N_H=550$, $N_L=2200$): https://i.imgur.com/ArqdQfF.png
>
> Burgers' ($N_H=700$, $N_L=2800$): https://i.imgur.com/HwzdET4.png
>
> BBM ($N_H=250$, $N_L=1000$): https://i.imgur.com/aiTf7PH.png
>
> BBM ($N_H=500$, $N_L=2000$): https://i.imgur.com/ZoBsDyB.png
>
> BBM ($N_H=750$, $N_L=3000$): https://i.imgur.com/cFDVjQ1.png
>
> BBM ($N_H=1000$, $N_L=4000$): https://i.imgur.com/QekAc8y.png
>
> Cahn-Hilliard ($N_H=250$, $N_L=1000$): https://i.imgur.com/1jyTN9t.png
>
> Cahn-Hilliard ($N_H=500$, $N_L=2000$): https://i.imgur.com/8D7bpEm.png
>
> Cahn-Hilliard ($N_H=750$, $N_L=3000$): https://i.imgur.com/BOu50c9.png
>
> Cahn-Hilliard ($N_H=1000$, $N_L=4000$): https://i.imgur.com/oSI7Unl.png

---

### Author Response · Authors · 2023-11-22
**Discussion reminder and summary of manuscript modifications**

We sincerely thank the reviewers for their invaluable feedback that helped us improve our paper. We have incorporated the suggested improvements and updated our manuscript.

The most significant modifications: we further clarified our motivation for low-resolution training (Introduction), justified choosing the LSTM architecture (Section 2.2), clarified the reasoning behind the training procedure (Section 2.4), corrected the reported standard deviations of the aggregated results (Table 1), and included comparisons to the state of the art (Section 4.3 and Figure 3). The description of training data was moved to the Appendix due to space considerations.

Please let us know if there are any remaining questions we can address or if there are any additional clarifications required.

---

### Meta-Review · Area_Chair_nP3E · 2023-12-06

**Metareview:**

This paper introduces a new model architecture, DON-LSTM, designed to effectively model multi-resolution data while accounting for temporal dependencies. The proposed architecture harnesses the strengths of both LSTM and DeepONets, resulting in enhanced performance for long-time system evolutions. All reviewers have acknowledged the significance of the problem addressed in this paper. However, there still exist strong concerns about the extent of performance improvement (Reviewer FcVP and qWGe), the absence of theoretical justification (Reviewer FcVP and qWGe), and questions about the contribution’s significance in the fusion of DeepONets and LSTM (Reviewer FcVP).  Considering the opinions from the reviewers, this paper still needs further revision and improvement.

**Justification For Why Not Higher Score:**

The paper did not receive a higher score primarily due to concerns about the extent of performance improvement of the DON-LSTM model, the lack of theoretical justification for the model's design, and questions regarding the significance of its contribution in combining DeepONets and LSTM. These unresolved issues indicate that further revision and improvement are needed.

**Justification For Why Not Lower Score:**

N/A

---

### Decision · Program_Chairs · 2024-01-16

Reject